# Antenatal Iron-Folic Acid Supplementation Is Associated with Improved Linear Growth and Reduced Risk of Stunting or Severe Stunting in South Asian Children Less than Two Years of Age: A Pooled Analysis from Seven Countries

**DOI:** 10.3390/nu12092632

**Published:** 2020-08-28

**Authors:** Yasir Bin Nisar, Victor M. Aguayo, Sk Masum Billah, Michael J. Dibley

**Affiliations:** 1Department of Maternal, Newborn, Child and Adolescent Health, World Health Organization, 20 Avenue Appia 1211, 27 Geneva, Switzerland; nisary@who.int; 2Programme Division, Nutrition Programme, United Nations Children’s Fund (UNICEF), 3UN Plaza, New York, NY 10017, USA; vaguayo@unicef.org; 3Sydney School of Public Health, The University of Sydney, Sydney, NSW 2006, Australia; billah@icddrb.org

**Keywords:** iron and folic acid, antenatal supplements, birth size, length-for-age Z score, stunting, children under-two, South Asia

## Abstract

In South Asia, an estimated 38% of preschool-age children have stunted growth. We aimed to assess the effect of WHO-recommended antenatal iron, and folic acid (IFA) supplements on smaller than average birth size and stunting in South Asian children <2 years old. The sample was 96,512 mothers with their most recent birth within two years, from nationally representative surveys between 2005 and 2016 in seven South Asian countries. Primary outcomes were stunting [length-for-age Z-score (LAZ) < –2], severe stunting [length-for-age Z-score (LAZ) < –3], length-for-age Z score, and perceived smaller than average birth size. Exposure was the use of IFA supplements. We conducted analyses with Poisson, linear and logistic multivariate regression adjusted for the cluster survey design, and 14 potential confounders covering the country of the survey, socio-demographic factors, household economic status, maternal characteristics, and duration of respondent recall. The prevalence of stunting was 33%, severe stunting was 14%, and perceived smaller than average birth size was 22%. Use of antenatal IFA was associated with a reduced adjusted risk of being stunted by 8% (aRR 0.92, 95% CI 0.89, 0.95), of being severely stunted by 9% (aRR 0.91, 95% CI 0.86, 0.96) and of being smaller than average birth size by 14% (aRR 0.86, 95% CI 0.80, 0.91). The adjusted mean LAZ was significantly higher in children whose mothers used IFA supplements. Maternal use of IFA in the first four months gestation and consuming 120 or more supplements throughout pregnancy was associated with the largest reduction in risk of child stunting. Antenatal IFA supplementation was associated with a significantly reduced risk of stunting, severe stunting, and smaller than average perceived birth size and improved LAZ in young South Asian children. The early and sustained use of antenatal IFA has the potential to improve child growth outcomes in South Asia and other low-and-middle-income countries with high levels of iron deficiency in pregnancy.

## 1. Introduction

Undernutrition is responsible for 3.1 million child deaths/year or 45% of global child deaths [1]. Both low birth-weight (<2500 g) and stunting (height or length-for-age-Z score < –2) are major public health problems [1,2]. Child stunting reflects poor nutrition and frequent infections before and after birth and leads to poor cognitive, motor, and socio-economic development [3]. In 2017, an estimated 155 million children <5 years had stunted growth [4]. South Asia has the largest prevalence of stunted children, with an estimated 40% of children <5 years old (61 million) stunted [1]. Globally, 20 million children are born low birth-weight each year [2] with 55% (11 million) in South Asia, which comprises Afghanistan, Bangladesh, Bhutan, India, Maldives, Nepal, Pakistan, and Sri Lanka [2]. Preterm birth or intrauterine growth restriction are the primary causes of low birth weight [5].

Globally, one-fifth of pregnant women have iron-deficiency anemia during pregnancy [1]. A meta-analysis shows that anemia during the first or second trimester increases the risk of prematurity and low birth weight [6], whereas antenatal iron supplementation significantly reduces maternal anemia [6,7,8]. The World Health Organization (WHO) recommends 60 mg iron daily and 400 µg folic acid throughout pregnancy for settings with a high prevalence of anemia in pregnancy (>40%) [9]. In most South Asian countries, health facilities and community health workers distribute iron and folic acid supplements (IFA). However, coverage remains low (ranging from 25% in Afghanistan [10] to 75% in the Maldives [11]).

There is a positive association between IFA supplementation during pregnancy, and birth-weight [6,7,8]. There are no trials that have directly assessed the effects of the recommended antenatal IFA (60 mg of elemental iron) supplements on the risk of childhood stunting. However, a cluster randomised controlled trial in Bangladesh compared antenatal, multiple micronutrients with IFA supplementation on postnatal linear and ponderal growth and the prevalence of stunting [12]. The dose of iron in both these supplements was 27 mg of elemental iron. The infants of the women given multiple micronutrients compared to the IFA group had slightly higher length for age z scores at birth, one, three, and six months, and less stunting at one and three months, but there were no differences in growth at 12 and 24 months. Also, trials in China [13] and Nepal [14] evaluated the long term impact of standard IFA plus zinc and multiple micronutrient supplements on childhood growth but showed no significant reduction in the risk of stunting. A pooled analysis of the Nepal Demographic and Health Survey (DHS) found use of IFA supplements during pregnancy significantly reduced the risk of stunting by 14% [15]. However, the impact of standard IFA supplementation in South Asia and its potential contribution to preventing childhood stunting is not yet established. Ref. [16] We used nationally representative surveys to investigate the association between standard antenatal IFA supplementation and childhood stunting, severe stunting, length-for-age Z scores, and perceived birth size in South Asian children aged <2 years.

## 2. Materials and Methods

### 2.1. Data Sources and Study Design

Our analyses examined nationally representative survey data collected between 2005 and 2016 from seven South Asian countries. The data analysed included Demographic and Health Surveys (DHS) from Bangladesh, Maldives, Nepal and Pakistan, the National Family Health Survey (NFHS) from India, and National Nutrition Surveys (NNS) from Afghanistan and Bhutan. In each of these surveys, we used the information on the women’s most recent live birth within two years before the interview. We obtained de-identified DHS and NFHS (India) data from The Demographic and Health Surveys (DHS) Program website (http://dhsprogram.com/) and for NNS (Afghanistan and Bhutan) from the institutions that conducted the surveys. Table 1 shows the country-wise source of data, type of survey, year of survey, and data availability. Each survey included a nationally representative sample of households selected by multi-stage cluster probability sampling, and collected information on socio-demographic, health, and nutrition indicators [10,17,18]. The survey procedures have been described in detail [10,11,18,19,20,21,22]. Briefly, households in each predefined area were randomly selected, and in each selected household, all ever-married women 15–49 years (reproductive age) were interviewed [10,11,18,19,20,21,22]. The respondents reported information about the birth history, live-births in chronologic order, birth date, singleton or multiple births, and sex of the child [10,11,18,19,20,21,22]. The surveys have records on the use of antenatal care services and anthropometric data for the most recent live-birth in the last five years [10,11,18,19,20,21,22].

We use birth history to construct cohorts from the most recent live birth within two years before the interview. Exposure to antenatal IFA supplementation was collected retrospectively based on maternal recall. We also used length-for-age in children less than two years and perceived birth size. To evaluate the primary outcome, we used cohorts of women of reproductive age and their most recent live birth. We examined a total of 96,512 (90,213 weighted) most recent live births two years before the interview, where anthropometric measurements were also available. We applied sample weights within each survey to adjust for multi-stage cluster sampling and selected the most recent live birth to limit recall bias, particularly for maternal use of antenatal care (ANC) services, including the use of IFA supplements.

### 2.2. Ethics

The 2007 Bangladesh DHS and the 2006 Nepal DHS were approved by The Institutional Review Board of Macro International) (Macro project number 31406.00.002.12). The NFHS-4 India Survey of 2015/16 was approved by the Institutional Review Board of ICF International (ICF project number 631561.0.000.00.071.01). The 2010 Maldives DHS and 2012–13 Pakistan DHS were approved by the Institutional Review Board of ICF (ICF project number 31561.00.000.00). The 2015 Bhutan National Nutrition and Anaemia Survey, obtained ethical clearance from the Bhutan Research Ethics Board of Health (no identification number available). And the 2013 Afghanistan National Nutrition Survey, was approved by the Institutional Review Board of the Ministry of Public Health, Government of the Islamic Republic of Afghanistan (no identification number available). In all the surveys, the respondents provided informed consent.

### 2.3. Study Outcomes

Our primary outcome was low length-for-age (stunting) of the child measured at the interview, which was the difference between the measured length and the mean length in the same age/sex group of WHO 2006 child growth standards [16] expressed as the number of standard deviations or Z-scores. Children were classified as stunted if their length-for-age Z-score (LAZ) was < –2 Z-scores below the age- and sex-specific mean length value from the WHO Child Growth Standard. Other study outcomes were the length-for-age Z score expressed as a continuous variable, severe stunting (LAZ < –3 Z scores), and perceived birth size based on maternal recall and coded ‘larger than or equal to average birth size’ and ‘smaller than average birth size’. The perception of birth size was recorded only in the Maldives, Nepal, Pakistan, and India.

### 2.4. Exposure Variables and Potential Confounding Factors

The primary exposure was the maternal use of IFA during the last pregnancy within two years before the interview. We classified consuming IFA if the mother reported taking supplements for at least a day during pregnancy. We assessed the effect of the total number of IFA supplements consumed where the survey recorded this information. We excluded records where the mother reported the use of >240 supplements. We also excluded records where the reported number of supplements consumed was higher than the maximum feasible number. Assuming consumption is once daily, we excluded the record if a woman reported consuming more than 150, 120, 90, 60, or 30 supplements when she started in the fifth, sixth, seventh, eighth, or ninth month of gestation, respectively. We used the time of first ANC examination as a surrogate for the time of initiation of IFA and categorised it as before or after four months of pregnancy. We also constructed a variable combining the timing of initiation and the total number of supplements consumed.

We assessed 14 potential confounding factors, i.e., community-level and socio-economic status, maternal and child characteristics, and perinatal health care services that were available in the surveys from all seven countries (Table 2).

We included all 14 potential confounders in the final models of the effect of the use of IFA on birth size and stunting. The surveys from Bangladesh and Bhutan did not collect information about the number of IFA supplements used during pregnancy. Hence these survey data were not included in the analysis of the number of IFA supplements. In sub-analyses of the data from India, we also assessed short maternal stature and child dietary diversity as potential confounding factors. We assessed for collinearity between these potential confounding factors. We constructed a household wealth index for economic status using pooled survey data and principal component analysis [23] of an inventory of household facilities and assets including the type of toilet, the main floor material, the source of drinking water, the availability of electricity, the possession of radios, televisions, refrigerators, telephones or bicycles. Using this index, we ranked households across all the surveys and then divided them into quintiles. Also, we adjusted our results for the duration of the recall, which is the time between the date of the childbirth and the date of the interview.

### 2.5. Statistical Analysis

We used ‘svy’ commands from Stata 14 (Stata-Corp, College Station, TX, USA) for all analyses to adjust for the cluster sampling design and apply the sampling weights in each survey. We calculated frequencies for study factors using the pooled data with weighted percentages and 95% confidence intervals (95% CI). Using country-specific data, as well as pooled data from all surveys, we performed Poisson regression for the stunting (LAZ < –2 Z score) and severe stunting (LAZ < –3 Z score) outcomes; linear regression for length-for-age Z scores as a continuous variable; and logistic regression for smaller than average perceived birth size. Initially, we fitted univariate regression models for each potential factor and followed by a multi-stage regression model. At the first stage, we included all community-level and socio-economic status variables, and we removed the non-significant factors (*p* > 0.05) using backward elimination. In the second stage, we included maternal and child characteristics. In the third stage, we assessed the number of antenatal care visits with the significantly associated factors from the previous stages. At the last stage, exposure variables were included separately with other significantly associated factors from stage 3. We used a significance level of <0.05 except for the duration of recall, which we retained in all models. To adjust models for the perception of birth size, we used data only from the four countries with this information.

## 3. Results

### 3.1. Prevalence of the Background Characteristics, Exposure and Outcome Variables

#### 3.1.1. Background Characteristics

Table 3 presents the background characteristics. Approximately 86% of the women in the sample were from India, with less than 1% from Bhutan.

Nearly three quarters (73%) were from rural areas, and about one third (32%) had no education. About 61% of households used biomass cooking fuel, nearly half (48%) had improved home sanitation facilities, and over four-fifths (83%) reported improved drinking water use. About one-third of mothers (34%) were teenagers at the time of giving birth. One in six (16%) children experienced diarrhea during the two weeks before the interview. Nearly one in five mothers (19%) reported no antenatal care visits during their recent pregnancy.

#### 3.1.2. Exposure and Outcome Variables

Table 4 shows the prevalence of the exposure and outcome variables. About 27% (95% CI 26.0%, 27.0%) of mothers did not use IFA during their recent pregnancy; slightly over two-fifths (43%) reported consumption of <120 IFA supplements and one in eight mothers (13%) reported consumption of ≥120 IFA supplements.

Over half of the mothers (55%) reported the initiation of supplements before the first four months of pregnancy. Nearly 60% of the mothers who consumed ≥120 supplements throughout pregnancy reported initiation of the supplements before the fourth month of pregnancy. Stunting was present in 29% (91%CI 28.3%, 29.2%) of children and severe stunting in 11% (95% CI 10.9%, 11.5%). The rate of stunting ranged from 15% in Bhutan to 34% in Afghanistan.

### 3.2. Factors Associated with Stunting and Length-for-Age

The factors associated with stunting in the children from the last pregnancy within two years before the interview (Table 5) included, mothers who had up to a primary level of education, used biomass fuel for cooking, had unimproved sanitary facilities, belonged to the poorest household wealth quintile, children of older age, and children experiencing diarrhea during the two weeks before the interview. Children whose mothers were aged 25 years or more or had ANC visits had significantly less risk of stunting.

Table 6 presents the factors associated with severe stunting in the children from the last pregnancy within two years before the interview. These factors included selected countries, mothers who had primary level or no education, use of biomass fuel for cooking, unimproved sanitary facilities, belonged to the poorer households, male children, children of older age, and children whose mothers had ANC visits.

Table 7 presents the factors associated with mean length-for-age Z scores, which included selected countries, use of biomass fuel for cooking, mothers who had a primary level or no education, belonged to the poorer households, unimproved source of drinking water, unimproved sanitary facilities, older and younger women, male children, children of older age, children with recent diarrhea, and children whose mothers had four or more ANC visits.

Perceived smaller than average birth size was associated with mothers who had no education, used biomass fuel for cooking, belonged to the poorest household wealth quintile, had height <145 cm, had multiple births, had female babies or were first-time mothers (data not shown).

### 3.3. Effect of IFA on Stunting and Severe Stunting

Figure 1 shows the effect of IFA on child stunting as a forest plot. The adjusted relative risk of being stunted was 8% significantly lower in children whose mother took any IFA during pregnancy (aRR 0.92, 95% CI 0.89, 0.95). Children whose mother initiated the supplements during the first four months of pregnancy had a 10% significantly lower adjusted relative risk of stunting (aRR 0.90, 95% CI 0.87, 0.93); and children whose mother used ≥120 supplements during pregnancy had a 14% significantly lower adjusted relative risk of stunting (aRR 0.86, 95% CI 0.81, 0.91). When mothers initiated supplementation during the first four months and took ≥120 supplements during pregnancy, the child’s adjusted relative risk of being stunted was 14% lower (aRR 0.86, 95% CI 0.81, 0.91). Country-wise analyses showed that any antenatal IFA reduced the adjusted risk of stunting in all countries, but the reduced risk was only statistically significant in India, Pakistan, and Nepal (Appendix A). We re-analysed the data from India, which had information on maternal height as a potential confounding factor. Maternal short stature (height <145 cm) was associated with an increased risk of stunting (aRR 1.47, 95% CI 1.42, 1.53). Higher child dietary diversity score (number of food groups out of seven consumed in the 24 h prior to interview) was associated with a decreased risk of stunting (aRR 0.98, 95% CI 0.97, 0.99). After the inclusion of these factors in the multivariable model any IFA supplementation remained protective for the risk of stunting in India (aRR 0.96, 95% CI 0.92, 0.99) but with a slightly decreased effect (see Appendix A).

The adjusted risk of being severely stunted was 9% lower (aRR 0.91, 95% CI 0.86, 0.96) in children whose mother used IFA during pregnancy (Figure 2). The adjusted relative risk of severe child stunting was 12% lower (aRR 0.88, 95% CI 0.80, 0.98) among the children of women who initiated IFA during the first four months of pregnancy, and who consumed ≥120 supplements (aRR 0.88, 95% CI 0.80, 0.98).

### 3.4. Effect of IFA on Length-for-Age

Table 8 presents the results of the adjusted linear regression for LAZ. The adjusted mean LAZ was significantly higher in children whose mother used IFA (*p* < 0.0001). The adjusted mean LAZ of children was highest among the women who consumed ≥120 supplements irrespective of the timing of initiation of the IFA supplements.

### 3.5. Effect of IFA on Birth Size

Figure 3 shows that the adjusted risk of perceived smaller than average birth size was reduced by 14% (aRR 0.86, 95% CI 0.80, 0.91) with maternal antenatal IFA and was significantly reduced by 22% (aRR 0.78, 95% CI 0.70, 0.87) in children whose mother initiated IFA during the first four months of pregnancy and took ≥120 supplements during pregnancy. We adjusted the multivariate models for the maternal perception of birth size in the four countries which collected this information (n = 86,103). Appendix A shows the factors associated with stunting. The adjusted risk of stunting was significantly higher by 31% (aRR 1.31, 95% CI 1.27, 1.36) and severe stunting by 41% (aRR 1.41, 95% CI 1.32, 1.51) in children who had smaller than average birth size. Appendix A shows the effect of antenatal IFA in the same models with perceived birth size in which the adjusted relative risk of being stunted was 7% lower in children whose mother took antenatal IFA (aRR 0.93, 95% CI 0.90, 0.96). Similarly, when mothers initiated entation during the first four months of pregnancy and took ≥120 supplements during pregnancy, the children’s adjusted relative risk of being stunted was 13% lower (aRR 0.87, 95% CI 0.82, 0.93).

## 4. Discussion

### 4.1. Main Findings and Their Significance

In this analysis of seven countries in South Asia, antenatal IFA reduced the risk of stunting in children <2 years by 8% and of being severely stunted by 9%. These results were adjusted for the country of survey, maternal educational status, fuel used for cooking, source of drinking water, sanitation facilities, household wealth index, sex of child, recent child diarrhea, and in a subsample of the Indian data, maternal stature and recent dietary diversity. Consumption of ≥120 supplements reduced the adjusted risk of stunting by 14%. When women initiated antenatal IFA during the first four months of pregnancy, the adjusted risk of being stunted was reduced by 10%. With the initiation of IFA supplements during the first four months of pregnancy and the use of ≥120 supplements during pregnancy, the adjusted risk of stunting was reduced by 14%, and there was a significantly higher adjusted mean LAZ. In sub-analyses, we observed similar preventive effects of antenatal IFA after adjusting for perceived birth size.

Our study is the first adequately powered analysis to report a protective association between antenatal IFA and child stunting in South Asia. In most of the countries in our analysis, IFA was initiated after four months of gestation, whereas we found a greater preventive effect of IFA on stunting when supplements started during the first four months and with the use of ≥120 supplements during pregnancy. Similarly, we have previously reported an effect of early initiation and consumption of a higher number of IFA on neonatal and child mortality in Nepal, Pakistan, and Indonesia [24,25,26,27]. A multi-country prospective randomized controlled stepped wedge trial with long term follow-up is now needed to confirm the effect of IFA supplements on the survival and growth of infants.

### 4.2. Comparison with Other Studies

Our findings are consistent with a pooled analysis from three Nepal DHS (2001, 2006 and 2011), which found stunting in 31% of children under two. Maternal use of IFA supplements during pregnancy significantly reduced the risk of stunting by 14%. Moreover, in Nepalese children whose mother-initiated supplements in the first six months and consumed 90 or more supplements during pregnancy, there was a higher adjusted risk reduction (23%). The adjusted risk of severe stunting in Nepalese children <2 years was significantly reduced by 37% with IFA during pregnancy. Similarly, with antenatal IFA, the adjusted risk of smaller than average birth size was significantly reduced by 19% [15], and a secondary analysis of Pakistan DHS data reported that the adjusted risk of maternal perceived smaller than average birth size was significantly reduced by 18% in children <5 years old whose mother used IFA during pregnancy. Moreover, the adjusted risk of maternal perceived smaller than average birth size was significantly reduced by 19% in children whose mothers began IFA in the first trimester [28].

Studies show that maternal perception of small birth size [29] and low birth-weight [30] increase the risk of stunting. Similarly, in a sub-analysis in the current study, there was an increased risk of stunting and severe stunting with perceived smaller than average birth size. A meta-analysis of 19 cohort studies (n = 44,374) from low- and middle-income countries found a significantly higher odds of stunting in children aged 12–60 months with low birth-weight (<2.5 kg within 72 h of birth), small for gestational age (<10th percentile) and prematurity (gestational age <37 weeks). Further, small for gestational age and term infants with low birth-weight had a higher odds of stunting (OR 3.00, 95% CI 2.36, 3.81) compared to small for gestational age and term infants without low birth-weight (OR 1.92, 95% CI 1.75, 2.11) [30].

Earlier meta-analyses examining the effects of antenatal IFA report a 19%–20% reduction in low birth-weight with the use of IFA in pregnancy [6,7,8]. A Cochrane review found a significant risk reduction in low birth-weight by 12% and in small for gestational age infants by 8% with multiple micronutrients compared to IFA. [31]. Another recent meta-analysis of individual patient data from 17 randomized controlled trials [32] in LMICs found that antenatal multiple micronutrient supplements compared to IFA alone reduced low birth weight by 19% (RR 0.81, 95% CI 0.74–0.89; *p* = 0.049), and small-for-gestational-age births by 8% (0.92, 0.87–0.97; *p* = 0.030). The more substantial reduction in the risk of low birth-weight and small-for-gestational-age births by multiple micronutrients compared to IFA alone could derive from enhanced absorption of iron due to the presence of other micronutrients such as ascorbic acid [33]. Despite the improvements in birthweight from multiple micronutrient supplements compared to IFA alone, a meta-analysis of six trials with anthropometric follow-up found no differences in the children’s weight-for-age or height-for-age Z scores among children aged 12 to 30 months [34]. A more recent large scale trial in Bangladesh on the effects of antenatal multiple micronutrient supplementation on growth found improvements in length-for-age Z scores and a 9% reduction in stunting at three months of age but not in older children [12]. To adequately interpret the literature comparing the effects of multiple micronutrients and IFA in pregnancy, we need to understand the effects of IFA alone.

In contrast to the preventive effect of IFA in the current study, two cluster randomized controlled trials from China [13] and Nepal [14] have shown no significant risk reduction of stunting from maternal IFA use, multi-micronutrients or micronutrient supplements. The lack of a measured effect of IFA could have been due to inadequate statistical power, the low prevalence of low birth weight, and the low rate of stunting [13,14].

### 4.3. Program Implications

We found that young South Asian children whose mother initiated IFA within the first four months of gestation and consumed 120 or more supplements during pregnancy had a 22% lower adjusted risk of having a smaller than average birth size, a 14% lower risk of being stunted, and a 12% lower risk of being severely stunted. These are significant findings for policymakers, program managers, and stakeholders in the fields of maternal, newborn, and child health and nutrition to support the development and implementation of programs for early initiation of IFA during pregnancy. Currently, most antenatal programs in South Asia initiate IFA supplements during the second trimester of pregnancy. Further, the coverage of supplementation is low, indicating a need to improve timely initiation of, adherence to, and coverage of IFA programs.

A good example is Nepal, where a district-level intervention package to improve IFA coverage was implemented across the country and successfully improved the coverage and adherence to antenatal IFA [35]. However, before designing interventions, it is vital to explore the country or context-specific barriers to coverage, early initiation, and adherence to IFA. Recent studies from Pakistan have identified that the main factors for non-use of IFA were: forgetting to take the supplements; non-availability of IFA supplements; limited financial capacity; lack of antenatal care services; family members not allowing supplement use; not knowing the benefits or no education; fear or experience of side effects; considering them contraceptives; and felt better thus stopped [24,36].

### 4.4. Strengths and Limitations

The strengths of this study include the analysis of pooled data from nationally representative surveys that provided a large sample of mother-child dyads, where the timing of the recalled exposure was antecedent to the outcome. Secondly, the study had >90% power to show significant public health differences in the risk of stunting with the use of IFA. Thirdly, to increase the validity, we used multivariate models to adjust for potential confounding factors at the community, household, and individual levels. Lastly, we selected live births two years before the interview to minimize recall bias [37,38,39] and adjusted our models for the duration of the recall period.

The study has three key limitations. Firstly, we did not assess the biochemical indicators of iron status. Secondly, there may be misclassification from the maternal recall of the number of IFA consumed during pregnancy with heaping of our data at intervals of 30. Also the women may have recalled other types of supplements. But this error might be uncommon because enumerators in these surveys showed samples of local iron and folic acid tablets to assist with recall of the correct type of supplements. Although there was a concurrent collection of information on exposures and outcomes, the retrospective measurement of IFA supplementation before birth and current measurement of stunting strengthens possible causal inferences. It is unlikely that the child’s birth size or stunting status influenced the recall of antenatal IFA use since the data comes from multi-component surveys covering women’s reproductive and child health, as well as other demographic and health information. Moreover, the outcome status of the child was not known in the interview because the anthropometric data required subsequent data processing. Thirdly, there is also a possibility of remaining confounding despite our adjustment for a large number of potential confounding factors, because mothers were not randomly assigned to IFA. Also, several potential confounders were not available in the data sets, including the type of complementary feeds and the mother’s health-seeking, caring and compliance behaviors. Women who used IFA might also be more aware and adherent to other preventive health behaviors that might contribute to lower rates of child stunting.

## 5. Conclusions

We found that the use of antenatal IFA was significantly associated with a reduced risk of smaller than average birth size, being stunted, and being severely stunted in South Asian children aged <2 years. The greatest positive effect was observed with early initiation of IFA supplementation at or before four months and consuming ≥120 IFA supplements throughout pregnancy, suggesting a timing- and dose-dependent response. Our findings support the need to more effectively implement at scale antenatal IFA supplementation, particularly in settings, like South Asia, where there is a high prevalence of anemia in pregnancy, LBW and stunting.

## Figures and Tables

**Figure 1 nutrients-12-02632-f001:**
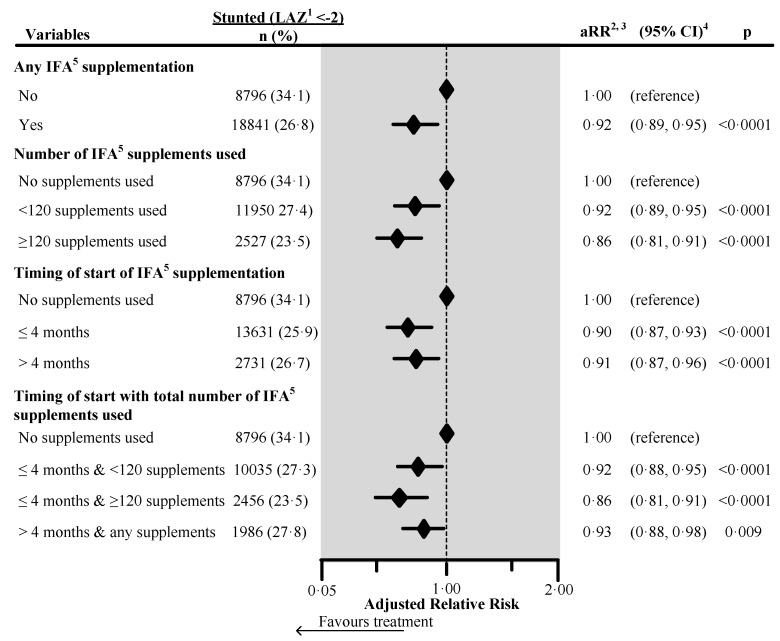
Effect of any iron/folic acid (IFA) supplementation, the number of supplements used, the timing of the start of supplements, and a combination of the timing of start with the number of supplements used on child stunting in South Asia, using adjusted Poisson regression. The surveys from Bangladesh and Bhutan did not collect information about the number of IFA supplements used during pregnancy. Hence these survey data were not included in the number of IFA supplements analysis (n = 4744). Also, we excluded records of mothers from analysis who reported consumption of >240 IFA supplements during pregnancy (n = 85). We also excluded those records where the number of supplements consumed exceeded the number possible when supplementation started between the fifth and ninth months of pregnancy (n = 183). ^1^ Length-for-Age Z-score. ^2^ Adjusted for country, area of residence, maternal marital status, maternal educational status, fuel used for cooking, source of drinking water, sanitation facilities, pooled household wealth index, maternal age at childbirth, sex of child, the timing of initiation of breastfeeding, age of the child, and child had diarrhea during last two weeks before the interview. Also, we adjusted the model for the duration of recall. ^3^ aRR: Adjusted relative risk. ^4^ CI: Confidence interval. ^5^ IFA: Iron/folic acid.

**Figure 2 nutrients-12-02632-f002:**
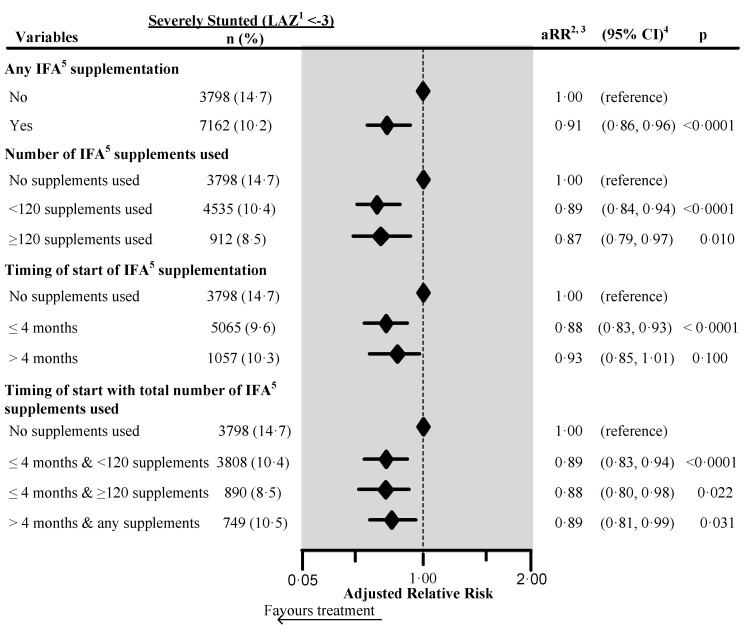
Effect of any iron/folic acid (IFA) supplementation, the number of supplements used, the timing of the start of supplements, and a combination of the timing of start with the number of supplements used on child severe stunting in South Asia, using adjusted Poisson regression. The surveys from Bangladesh and Bhutan did not collect information about the number of IFA supplements used during pregnancy. Hence these survey data were not included in the number of IFA supplementation analyses (n = 4744). Also, we excluded records of mothers from analysis who reported consumption of >240 IFA supplements during pregnancy (n = 85). We also excluded those records where the number of supplements consumed exceeded the number possible when supplementation started between the fifth and ninth months of pregnancy (n = 183). ^1^ Length-for-Age Z-score. ^2^ Adjusted for country, area of residence, maternal marital status, maternal educational status, fuel used for cooking, source of drinking water, sanitation facilities, pooled household wealth index, maternal age at childbirth, sex of child, the timing of initiation of breastfeeding, age of the child, and child had diarrhea during last two weeks before the interview. Also, we adjusted the model for the duration of recall. ^3^ aRR: Adjusted relative risk. ^4^ CI: Confidence interval. ^5^ IFA: Iron/folic acid.

**Figure 3 nutrients-12-02632-f003:**
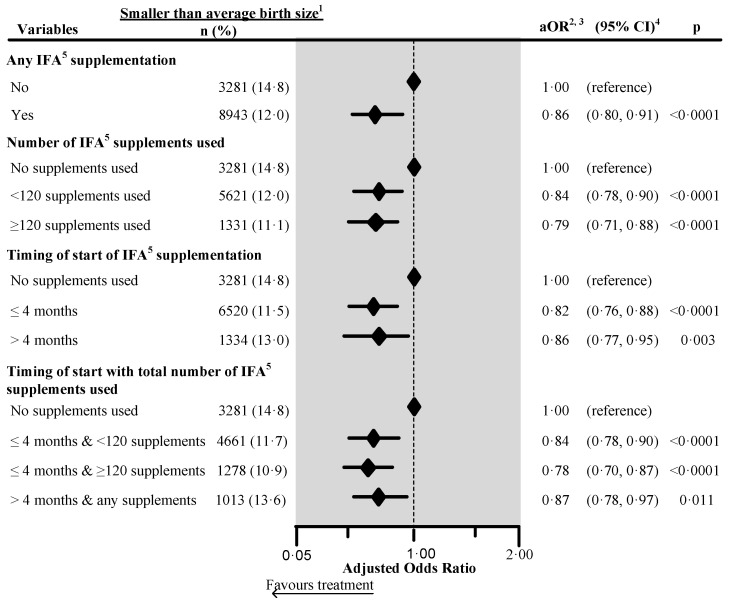
Effect of any iron/folic acid (IFA) supplementation, the number of supplements used, the timing of the start of the supplements and a combination of the timing of start with the number of supplements used on the maternally perceived birth size in four South Asian countries ^6^, using adjusted logistic regression. ^1^ Included smaller than average and smallest categories. ^2^ Adjusted for country, place of residence, maternal marital status, maternal educational status, maternal working status, paternal educational status, paternal working status, fuel used for cooking, source of drinking water, sanitation facilities, pooled household wealth index, maternal age at childbirth, maternal desire for pregnancy, maternal smoking status, maternal height, birth outcome status, birth interval and rank, and sex of child. Also, we adjusted the model for the duration of recall. ^3^ aOR: Adjusted odds ratio. ^4^ CI: Confidence interval. ^5^ IFA: Iron/folic acid. ^6^ The surveys from India, Maldives, Nepal, and Pakistan collected information on maternally perceived birth size. We excluded records of mothers from analysis who reported consumption of >240 IFA supplements during pregnancy. We also excluded those records where the number of supplements consumed exceeded the number possible when supplementation started between the fifth and ninth months of pregnancy.

**Table 1 nutrients-12-02632-t001:** Country-wise type of survey, year of survey and data source for selected country-specific survey data used in the current study.

Country	Type & Year of Survey	Data Source
Afghanistan	National Nutrition Survey (NNS) 2013	Institutional (UNICEF)
Bangladesh	Demographic and Health Survey (DHS) 2007	Public domain
Bhutan	National Nutrition and Anaemia Survey 2015	Institutional (UNICEF)
India	National Family Health Survey (NFHS) 2015/16	Public domain
Maldives	Demographic and Health Survey (DHS) 2010	Public domain
Nepal	Demographic and Health Survey (DHS) 2006	Public domain
Pakistan	Demographic and Health Survey (DHS) 2012/13	Public domain

**Table 2 nutrients-12-02632-t002:** Operational definition and categorisation of the potential confounding variables used in the analysis.

Variables	Definition and Categorisation
**Community-level and socio-economic factors**	
Country	Country of survey (1 = India; 2 = Afghanistan; 3 = Bangladesh; 4 = Bhutan; 5 = Maldives; 6 = Nepal; 7 = Pakistan)
Place of residence	Place of residence of the respondent (1 = urban; and 2 = rural)
Maternal marital status	Marital status of the mother (1 = currently married, and 2 = formerly married)
Maternal educational status	Maternal level of attained education (1 = secondary and above; 2 = completed primary; and 3 = no education)
Fuel used for cooking	Fuel used for cooking at home (1 = biomass energy; and 2 = natural gas).
Source of water used for drinking	Source of water used for drinking at home was classified based on WHO/UNICEF guidelines (1 = improved; and 2 = unimproved).
Sanitation facility	Sanitation referred to the toilet facility at home and was classified based on WHO/UNICEF guidelines (1 = improved; and 2 = unimproved).
Pooled household wealth index	A composite index of household amenities using pooled data and a principal component analysis [23] of household assets that was ranked into quintiles
**Maternal and child characteristics**	
Maternal age at childbirth	Maternal age at childbirth (1 = 20–24 years; 2 = <20 years; and 3 = ≥25 years).
Sex of child	Sex of the child (1 = female; and 2 = male).
Age of child	Age of the child in months (as a continuous variable)
Initiation of breastfeeding	Timing of initiation of breastfeeding (1 = <1 h; 2 = 1 to 24 h; 3 = >24 h; and 4 = never).
Child diarrhea in the last two weeks	The child had diarrhea within two weeks before the interview date (1 = no; and 2 = yes).
**Perinatal health services variables**	
Number of ANC visits	Number of antenatal care visits (1 = no ANC visit; 2 = <4 visits; and 3 = ≥4 visits).

ANC: antenatal care.

**Table 3 nutrients-12-02632-t003:** Prevalence of community-level, socio-economic status, maternal and child characteristics and perinatal health services of most recent live-births two years before the interview in pooled data from seven countries in South Asia (N = 96,552).

Variables	Unweighted	Weighted ^†^	
	N	n	%
**Community-level and socio-economic status factors**			
**Country**			
India	83,594	77,276	85.7
Afghanistan	6343	6366	7.1
Bangladesh	2079	2058	2.3
Bhutan	562	526	0.6
Maldives	992	977	1.1
Nepal	1944	1841	2.0
Pakistan	1038	1170	1.3
**Place of residence**			
Urban	22,165	24,163	26.8
Rural	74,387	66,051	73.2
**Maternal marital status**			
Currently married	95,452	89,480	99.2
Formerly married	1100	733	0.8
**Maternal educational status**			
Secondary and above	51,431	48,585	53.9
Up to primary	14,032	12,903	14.3
No education	31,068	28,710	31.8
Missing	21	16	<0.1
**Fuel used for cooking**			
Biomass energy	63,694	54,876	60.8
Natural gas	32,849	35,324	39.2
Missing	9	13	<0.1
**Source of drinking water**			
Improved	77,924	75,321	83.5
Unimproved	18,233	14,515	16.1
Missing	395	377	0.4
**Sanitary facility used at home**			
Improved	48,924	43,073	47.8
Unimproved	47,597	47,107	52.2
Missing	31	34	<0.1
**Household wealth index**			
Quintile 1 (Wealthiest)	12,423	13,005	14.4
Quintile 2	18,692	19,540	21.7
Quintile 3 (Middle)	18,502	17,222	19.1
Quintile 4	20,572	16,712	18.5
Quintile 5 (Poorest)	25,945	23,331	25.9
Missing	418	405	0.4
**Maternal and child characteristics**			
**Maternal age at childbirth**			
20–24 years	46,627	43,590	48.3
<20 years	31,610	30,879	34.2
>24 years	18,315	15,745	17.5
**Sex of Child**			
Male	50,347	47,093	52.2
Female	46,205	43,120	47.8
**Time to initiate breastfeeding after birth**			
Never breastfed	2834	2586	2.9
Within 1 h	65,016	59,355	65.8
1 h to 1 day	16,260	15,375	17.0
More than 1 day	12,418	12,878	14.3
Missing	24	19	<0.1
**Age of child (in months)**			
Mean (SD *)	11.6 (0.02)	11.7 (0.03)	
**The child had diarrhea in the last two weeks**			
No	81,288	75,486	83.7
Yes	15,226	14,683	16.3
Missing	38	44	<0.1
**Perinatal health services variable**			
**Number of ANC visits**			
No ANC visit	18,858	16,873	18.7
1–3 ANC visits	34,551	30,101	33.4
4 or more ANC visits	43,094	43,173	47.9
Missing	49	66	<0.1

ANC: Antenatal care * Standard deviation † Weighting was applied to compensate for the multi-stage cluster sampling design.

**Table 4 nutrients-12-02632-t004:** Prevalence of study exposure and outcome variables of the most recent live-births two years before the interview, in South Asia, 2005–2016.

Variable	Unweighted	Weighted		
	N	n	%	95% CI
**Exposure variables**				
**IFA used**				
No	25,928	23,892	26.5	(26.0, 27.0)
Yes	70,586	66,277	73.5	(73.0, 74.0)
Missing	38	44	<0.1	(<0.0, <0.1)
**Number of IFA used during Pregnancy**				
No IFA	25,928	23,892	26.5	(26.0, 27.0)
<120 IFA	43,810	38,673	42.9	(42.3, 43.4)
≥120 IFA	10,809	11,695	13.0	(12.5, 13.4)
Missing	16,005	15,956	17.8	(17.3, 18.1)
**Timing of initiation of IFA supplements**				
No IFA used	25,928	23,892	26.5	(26.0, 27.0)
>4 months of pregnancy	52,891	49,240	54.6	(54.0, 55.2)
≤4 months of pregnancy	10,276	10,117	11.2	(10.9, 11.6)
Missing	7457	6964	7.7	(7.4, 8.0)
**Timing of initiation and number of IFA used**				
No IFA	25,928	23,892	26.5	(26.0, 27.0)
>4 mo of pregnancy and any IFA	36,936	32,144	35.6	(35.1, 36.2)
≤4 mo of pregnancy and <120 IFA	10,496	11,382	12.6	(12.2, 13.1)
≤4 mo of pregnancy and ≥120 IFA	7187	6843	7.6	(7.3, 7.9)
Missing	16,005	15,956	17.7	(17.3, 18.1)
**Outcome variables**				
**Stunting (LAZ < −2)**				
Normal	68,794	64,259	71.1	(70.8, 71.7)
Stunting	27,758	25,954	28.9	(28.3, 29.2)
**Severe stunting (LAZ < −3)**				
Normal	85,557	80,092	88.8	(88.5, 89.1)
Severe stunting	10,995	10,121	11.2	(10.9, 11.5)
**LAZ (as a continuous variable)**				
Mean (SD)	−1.10 (0.01)		−1.10 (0.01)	
**Maternal perceived birth size ^1^**				
Average or larger than average	75,313	70,521	87.6	(87.3, 88.0)
Smaller than average	10,790	9969	12.4	(12.0, 12.7)

^1^ The India, Maldives, Nepal, and Pakistan surveys collected information on maternal perceived birth size (N = 86,103).

**Table 5 nutrients-12-02632-t005:** Factors associated with stunting (LAZ < −2) in South Asian children <2 years old (2005–2016): Results of adjusted Poisson regression ^1^.

Factors	aRR ^2^	95% CI ^3^	*p*
Community-Level and Socio-Economic Status Factors			
**Country**			
India	1.00	(reference)	
Afghanistan	1.75	(1.56, 1.96)	<0.0001
Bangladesh	0.84	(0.77, 0.92)	<0.0001
Bhutan	0.88	(0.62, 1.27)	0.510
Maldives	0.89	(0.76, 1.04)	0.150
Nepal	0.85	(0.77, 0.94)	0.002
Pakistan	1.11	(0.98, 1.24)	0.086
**Fuel used for cooking**			
Biomass energy	1.00	(reference)	
Natural gas	0.87	(0.83, 0.90)	<0.00010
**Maternal educational status**			
Secondary and above	1.00	(reference)	
Up to primary	1.20	(1.15, 1.25)	<0.0001
No education	1.33	(1.28, 1.38)	<0.0001
**Source of drinking water**			
Improved	1.00	(reference)	
Unimproved	0.95	(0.91, 0.99)	0.017
**Sanitary facility**			
Improved	1.00	(reference)	
Unimproved	1.11	(1.06, 1.15)	<0.0001
**Household wealth index**			
Quintile 1 (Wealthiest)	1.00	(reference)	
Quintile 2	1.16	(1.09, 1.25)	<0.0001
Quintile 3 (Middle)	1.20	(1.12, 1.28)	<0.0001
Quintile 4	1.27	(1.18, 1.36)	<0.0001
Quintile 5 (Poorest)	1.30	(1.20, 1.39)	<0.0001
**Maternal and child characteristics**			
**Maternal age at childbirth**			
20–24 years	1.00	(reference)	
<20 years	1.02	(0.99, 1.05)	0.110
>24 years	0.92	(0.88, 0.96)	<0.0001
**Sex of child**			
Female	1.00	(reference)	
Male	1.09	(1.06, 1.12)	<0.0001
**Age of child (in months)**	1.09	(1.08, 1.09)	<0.0001
**The child had diarrhea in the last two weeks**			
No	1.00	(reference)	
Yes	1.08	(1.07, 1.08)	<0.0001
**Perinatal health services variable**			
**Number of ANC visits**			
No ANC visit	1.00	(reference)	
1–3 ANC visits	0.93	(0.90, 0.96)	<0.0001
4 or more ANC visits	0.84	(0.81, 0.88)	<0.0001

^1^ Adjusted for country, area of residence, maternal marital status, maternal educational status, fuel used for cooking, source of drinking water, sanitation facilities, pooled household wealth index, maternal age at childbirth, sex of child, the timing of initiation of breastfeeding, age of the child, and child had diarrhea during last two weeks before the interview. Also, we adjusted the model for the duration of recall. ^2^ aRR: Adjusted relative risk. ^3^ CI: Confidence interval.

**Table 6 nutrients-12-02632-t006:** Factors associated with severe stunting (LAZ < −3) in South Asian children < 2 years old (2005–2016): Results of adjusted Poisson regression ^1^.

Factors	aRR ^2^	95% CI ^3^	*p*
Community-Level and Socio-Economic Status Factors			
**Country**			
India	1.00	(reference)	
Afghanistan	2.28	(1.88, 2.76)	<0.0001
Bangladesh	0.55	(0.44, 0.69)	<0.0001
Bhutan	0.91	(0.52, 1.59)	0.734
Maldives	0.68	(0.48, 0.95)	0.024
Nepal	0.64	(0.53, 0.78)	<0.0001
Pakistan	1.25	(1.00, 1.55)	0.05
**Fuel used for cooking**			
Biomass energy	1.00	(reference)	
Natural gas	0.83	(0.77, 0.89)	<0.0001
**Maternal educational status**			
Secondary and above	1.00	(reference)	
Up to primary	1.25	(1.15, 1.35)	<0.0001
No education	1.50	(1.41, 1.59)	<0.0001
**Source of drinking water**			
Improved	1.00	(reference)	
Unimproved	0.90	(0.83, 0.97)	0.005
**Household wealth index**			
Quintile 1 (Wealthiest)	1.00	(reference)	
Quintile 2	1.21	(1.07, 1.36)	0.002
Quintile 3 (Middle)	1.32	(1.17, 1.48)	<0.0001
Quintile 4	1.47	(1.31, 1.64)	<0.0001
Quintile 5 (Poorest)	1.70	(1.52, 1.91)	<0.0001
**Maternal and child characteristics**			
**Maternal age at childbirth**			
20–24 years	1.00	(reference)	
<20 years	1.01	(1.00, 1.19)	0.043
>24 years	1.09	(0.90, 1.14)	0.874
**Sex of child**			
Female	1.00	(reference)	
Male	1.17	(1.12, 1.23)	<0.0001
**Age in child (in months)**	1.09	(1.09, 1.10)	<0.0001
**Perinatal health services variable**			
**Number of ANC visits**			
No ANC visit	1.00	(reference)	
1–3 ANC visits	0.88	(0.83, 0.93)	<0.0001
≥4 ANC visits	0.77	(0.72, 0.83)	<0.0001

^1^ Adjusted for country, area of residence, maternal marital status, maternal educational status, fuel used for cooking, source of drinking water, sanitation facilities, pooled household wealth index, maternal age at childbirth, sex of child, the timing of initiation of breastfeeding, age of the child, and child had diarrhea during last two weeks before the interview. Also, we adjusted the model for the duration of recall. ^2^ aRR: Adjusted relative risk. ^3^ CI: Confidence interval.

**Table 7 nutrients-12-02632-t007:** Factors associated with length-for-age Z score (LAZ) in South Asian children <2 years old (2005–2016): Results of adjusted linear regression ^1^.

Factors	Coefficient	95% CI ^2^	*p*
Community-Level and Socio-Economic Status Factors			
**Country**			
India	1.00	(reference)	
Afghanistan	−0.25	(−0.40, −0.10)	0.001
Bangladesh	−0.03	(−0.10, 0.04)	0.375
Bhutan	0.60	(0.32, 0.89)	<0.0001
Maldives	−0.11	(−0.23, 0.01)	0.051
Nepal	−0.02	(−0.11, 0.07)	0.615
Pakistan	−0.18	(−0.29, −0,06)	0.002
**Fuel used for cooking**			
Biomass energy	1.00	(reference)	
Natural gas	0.18	(0.14, 0.22)	<0.0001
**Maternal educational status**			
Secondary and above	1.00	(reference)	
Up to primary	−0.19	(−0.23, −0.15)	<0.0001
No education	−0.31	(−0.34, −0.27)	<0.0001
**Household wealth index**			
Quintile 1 (Wealthiest)	1.00	(reference)	
Quintile 2	−0.14	(−0.19, −0.08)	<0.0001
Quintile 3 (Middle)	−0.19	(−0.25, −0.13)	<0.0001
Quintile 4	−0.27	(−0.33, −0.20)	<0.0001
Quintile 5 (Poorest)	−0.31	(−0.38, −0.25)	<0.0001
**Source of drinking water**				
Improved	1.00	(reference)	
Unimproved	0.05	(0.05, 0.0)	0.031
**Sanitary facility**			
Improved	1.00	(reference)	
Unimproved	−0.08	(−0.12, −0.04)	<0.0001
**Maternal and child characteristics**			
**Maternal age at childbirth**			
20–24 years	1.00	(reference)	
<20 years	−0.04	(−0.08, −0.01)	0.004
>24 years	0.13	(0.08, 0.17)	<0.0001
**Sex of child**			
Female	1.00	(reference)	
Male	−0.13	(−0.16, −0.10)	<0.0001
**Age in child (in months)**	−0.10	(−0.10, −0.10)	<0.0001
**The child had diarrhea in the last two weeks**			
No	1.00	(reference)	
Yes	−0.06	(−0.10, −0.02)	0.002
**Perinatal health services variable**			
**Number of ANC visits**			
No ANC visit	1.00	(reference)	
1–3 ANC visits	0.04	(−0.01, 0.08)	0.080
4 or more ANC visits	0.15	(0.10, 0.19)	<0.0001

^1^ Adjusted for country, area of residence, maternal marital status, maternal educational status, fuel used for cooking, source of drinking water, sanitation facilities, pooled household wealth index, maternal age at childbirth, sex of child, the timing of initiation of breastfeeding, age of the child, and child had diarrhea during last two weeks before the interview. Also, we adjusted the model for the duration of recall. ^2^ CI: Confidence interval.

**Table 8 nutrients-12-02632-t008:** Effect of any iron/folic acid (IFA) supplementation, the number of supplements used, the timing of the start of supplements and a combination of the timing of start with the number of supplements used on the length-for-age Z score in south Asian children <2 years old, adjusted linear regression ^1^.

Variables	Coefficients	95% CI ^2^		*p*
**IFA ^3^ supplements used**				
No	−	(reference)		
Yes	0.10	(0.07, 0.13)		<0.0001
**Number of IFA3 supplements used during pregnancy ^4^**				
No IFA	−	(reference)		
<120 IFA	0.08	(0.05, 0.12)		<0.0001
>120 IFA	0.15	(0.10, 0.21)		<0.0001
**Timing of initiation of IFA ^3^ supplements**				
No IFA used	−	(reference)		
<4 months	0.11	(0.08, 0.15)		<0.0001
>4 months	0.08	(0.03, 0.13)		0.002
**Timing of initiation and number of IFA ^3^ supplements used ^4^**				
No IFA	−	(reference)		
<4 months and <120 IFA	0.09	(0.05, 0.13)		<0.0001
<4 months and ≥120 IFA	0.15	(0.10, 0.21)		<0.0001
>4 months and any IFA	0.07	(0.01, 0.12)		0.019

^1^ Adjusted for country, area of residence, maternal marital status, maternal educational status, fuel used for cooking, source of drinking water, sanitation facilities, pooled household wealth index, maternal age at childbirth, sex of child, timing of initiation of breastfeeding, age of child, and child had diarrhea during last two weeks before interview. Also, we adjusted the model for the duration of recall. ^2^ CI: Confidence interval. ^3^ IFA: Iron/folic acid. ^4^ The surveys from Bangladesh and Bhutan did not collect information about the number of IFA supplements used during pregnancy. Hence these survey data were not included in the number of IFA supplementation analyses (n = 4744). Also, we excluded records of mothers from analysis who reported consumption of >240 IFA supplements during pregnancy (n = 85). We also excluded those records where the number of supplements consumed exceeded the number possible when supplementation started between the fifth and ninth months of pregnancy (n = 183).

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
