# Peer review of "Antenatal Iron-Folic Acid Supplementation Is Associated with Improved Linear Growth and Reduced Risk of Stunting or Severe Stunting in South Asian Children Less than Two Years of Age: A Pooled Analysis from Seven Countries"

_nutrients, 2020, doi:10.3390/nu12092632_

Round 1

Reviewer 1 Report

A well presented research paper. Methodology is clear and results well interpreted. 

Author Response

We thank the reviewer for the positive and supportive comments.

Reviewer 2 Report

This pooled analysis is very interesting and give valuable results to confirm the need to keep promote IFA supplementation during pregnancy. IFA represents a safe and inexpensive preventive measure to improve both mother and child health. The data are detailed and the chosen method is appropriate.

I have some global remarks: 

  • Standardize terms: baby or child.
  • Additional analysis: a stratified analysis according to the child’s age (depending of the data distribution but the most appropriate might be before/after 6-months). Because IFA decreases the risk of SGA and LBW, the effect of the supplementation could be higher in earlier months (and maybe assocations found in this analysis are exclusively driven by this early assocation).

And more specific ones: 

Line 66 to 68: What are the results of these surveys?

Line 70: The impact of standard IFA on growth is “not understood” (but already established) or not yet established?

Line 81: “de-identified” => anonymized?

Line 120: add the WHO reference.

Line 132-134: “Assuming consumption is once daily, we excluded the record if a woman reported consuming more than 150, 120, 90, 60, or 30 supplements when she started in the fifth, sixth, seventh, eighth, or ninth month of gestation, respectively”. Folic acid and iron may be in two separated tablets. Exclusion of consumption of more than 1 tablet per day could induce a bias.

Line 153-166: Specify the method used for each outcome. For which outcome Poisson or linear or logistic regression have been used. The analysis on perception of birth size must detailed, it seems that maternal height, multiple births have been included in the model but these variables haven’t been described in maternal characteristics (panel 2).

Table 5: Reference is 0 (not 1), linear regression.

Line 260-265 and line 280-285: “The surveys from Bangladesh and Bhutan did not collect information about the number of IFA…” These explanations remain to the statistical analysis part.

Line 389-393: The results of the meta-analysis compared two groups among anemic pregnant women vs non-anemic pregnant women? or multiple micronutrients supplements vs IFA? Or it was four groups comparisons? The sentence is not clear. Please add the reference.

Line 398: At what age the weight-for-age and height-for-age?

Reviewer 3 Report

Thank you for this opportunity to review this important article.

Overall this is a very well written paper about a large multi-country study that addresses an important issue in South Asia. The authors investigated the association between maternal iron and folic acid supplements (IFA) in the ante-natal period and infant stunting (length for age Z score (LAZ)) (any:≤2 and severe ≤3), and perceived birth size (3 countries only) in Southern Asian children ≤ 2 years. They used nationally representative survey data (including data on socio-demographic, health and nutrition indicators) from 7 countries (Bangladesh, Maldives, Nepal, Pakistan, India, Afghanistan and Bhutan), collected between 2005-2016, details of which have already been published. The sample consisted of 96,512 most recent live births who were under two years with anthropometric measurements were available. Mothers were asked to recall details about ante-natal care and use of IFA supplements. The majority of participants were from India (86%) with > 2000 births in Afghanistan and Bangladesh with > births. The authors found that IFA can reduce stunting and particularly sever stunting, especially when supplements are commenced before 4 months with evidence of a dose response.  They conclude by recommending a multi-country RCT with long term follow-up to confirm the effects of ante-natal IFA.

I have very minor suggestions.

The study used maternal recall of the use of IFA. While the maximum recall time was relatively short (up to ~2.75 years), can you please comment on the reliability of this? Are mothers likely to have been taking other supplements, not IFA, which might have been misclassified as IFA. In the background section, several trials of micronutrients plus or minus IFA are mentioned. Are the authors aware of any trials being run in the same population at the same time and thus ’contaminated’ the findings? The WHO recommends 60mg iron/400mcg folic acid daily. Is this the dose likely to have been provided in each of the study settings, given it is mentioned that the dose of iron was only 27mg in previous studies?

Lines 265, 285 : Forest plot footnotes Typos Leight-for-Age Z score

Author Response

This manuscript is a resubmission of an earlier submission. The following is a list of the peer review reports and author responses from that submission.

Round 1

Reviewer 1 Report

In this paper the authors investigated the correlation between antenatal maternal use of IFA (iron and folic acid supplements) and birth-weight. In the South-Asia 38% of preschool age children have stunted growth and one-fifth of pregnant women have iron-deficiency anemia during pregnancy. Trials in Bangladesh e China have been demonstred  a positive correlation between IFA supplementation during pregnancy  and birth-weight.

The World Health Organization reccomend 60 mg iron daily and 400 mcg folic acid troughout pregnancy.

In this study the primary outcomes were stunting (lenght-for-age Z score <-2), severe strunting (Z-score <-3) and perceived smaller than average birth size. Exposure was the use of IFA supplements during the last pregnancy within two years before the interview. The author classified consuming IFA if the mother reported taking supplements consumed where the survey recorded this information, and the timing of initiation of IFA (before/after four months of pregnancy).

In this analysis antenatal IFA  reduced the risk of stunting in children <2 years by 8% and of being severely stunted by 9%. Consumption of >120 supplements reduced the adjusted risk of stunting by 14%. When women initiated antenatal IFA during the first four months of pregnancy the adjusted risk of being stunted was reduced by 10%. So in this study report a protective association between antenatal IFA and child stunting in South Asia.

Here we report the most important criticism in the review of the paper:

There are difference in the cluster of women, in particular the economic and cultural level of women accessing care is not comparable No laboratory data are reported regarding iron and hemoglobin levels The data analysis includes subjective opinions issued in the interviews The timing regarding the intake of IFA are very different in order to be compared

Reviewer 2 Report

General Comments: Well written with detailed explanation of methods used to analyse the data available from these national household surveys. A concern would be the number of assumptions made within the data manipulation for analysis. However, analysis of available data is comprehensive and recognises potential limitations

Abstract: Provides a clear and concise summary of the main points of the paper

Introduction: Clear background that presents relevant information related to the aim of the study

Methods: Ln 78-81: the way this is described is not completely clear; consider rewording. Ln 118-119: Comprehensive details on data manipulation, in places this gets a little difficult to follow for the reader – is there a way of presenting this more clearly using a flow diagram or similar?

Results: Clearly presented

Tables and Figures: Tables and figures are clear and well annotated.

Discussion: Well written and highlights the main findings against previous research.  Data on breast feeding duration and/or timing of complimentary feeding was not included and only mentioned as a limitation of the study. Was this information unavailable or just not included for some reason?

When referring to the limitations it might be better to state that the authors identify 3 key or significant limitations as there may be many more minor ones that have not been noted.